# Musculoskeletal Morphology and Joint Flexibility-Associated Functional Characteristics across Three Time Points during the Menstrual Cycle in Female Contemporary Dancers

**DOI:** 10.3390/jfmk9010038

**Published:** 2024-02-25

**Authors:** Bárbara Pessali-Marques, Adrian M. Burden, Christopher I. Morse, Gladys L. Onambélé-Pearson

**Affiliations:** 1Department of Sport and Exercise Sciences, Institute of Sport, Faculty of Science and Engineering, Manchester Metropolitan University, Manchester M1 7EL, UK; barbarabastidores@gmail.com (B.P.-M.); a.burden@mmu.ac.uk (A.M.B.); c.morse@mmu.ac.uk (C.I.M.); 2Bastidores—Dance, Research and Training, Palhano, Brumadinho 35460000, Brazil

**Keywords:** dancer, flexibility, jumps, menstrual cycle, skeletal muscle, oestrogen, progesterone, relaxin

## Abstract

Findings are inconsistent with regards to whether menstrual cycle phase-associated changes in physical functioning exist. It is possible that such discrepancies are due to varying rigour in experimental approaches. The current study aimed to systematically evaluate any effect of carefully tracked menstrual cycle phase on precisely measured muscle structure and function in a physically active group (contemporary dancers). Eleven women aged (M [SD]) 23.5 [2.94] years, undergoing 10.5 [1.73] hours of contemporary dance practice and 6.12 [2.36] hours of other physical activity per week, were recruited. Sex hormone level (enzyme-linked immunosorbent assays (ELISA), skin temperature and ovulation kits), physical pain assessments (Ice Water Test, Visual Analogue Scale, The Physical Activity Readiness Questionnaire, Self-Estimated Functional Inability Because of Pain Questionnaire, and Pain Anxiety Symptoms Scale), muscle architecture measurement (B-mode ultrasonography), and physical functioning (dynamometry, force-platform and electromyography) on both lower limbs were measured at three time points during one cycle, following three months of menstrual cycle monitoring. There was no difference in musculoskeletal flexibility variables between follicular, ovulatory, or luteal phases. Nonetheless, oestrogen change was associated with variability in 11 musculoskeletal variables, progesterone change was associated with variability in 7, and relaxin change was associated with variability in 15. Negative correlations existed between progesterone and flexibility and between oestrogen and jump variables. Moreover, oestrogen and relaxin were associated with increased musculoskeletal compliance, whilst progesterone was associated with increased muscle stiffness. In short, in absolute sex hormone levels, ‘inter-individual’ variances appear more impactful than ‘intra-individual’ variances. Not only are oestrogen and progesterone associated with differing musculoskeletal outcomes, but relaxin is also associated with musculoskeletal compliance changes. These effects are anticipated to impact jump height and flexibility, and hence, they could be expected to affect overall physical performance, including dance.

## 1. Introduction

Oestrogen and progesterone fluctuations across menstrual cycle phases (MCPs) [1] have been found to lead to differing physiological effects [2,3] that may influence exercise performance [4,5,6]. Over the MCP, oestrogen is expected to peak twice (just after ovulation and then mid-follicular phase), whereas progesterone is expected to peak once (during the luteal phase). Relaxin, on the other hand, works to increase the compliance of some tendons [7] and is expected to rise after ovulation and drop again if pregnancy does not occur. Notably, oestrogen receptors are found in collagenous tissue such as the human anterior cruciate ligament [8]. Previous authors Park et al. [9] assessed knee laxity and its increase from the follicular phase to ovulation, linking these effects to high oestradiol during ovulation and to high progesterone during the luteal phase, though with marked inter-individual variability. Whilst some studies find greater joint laxity in the ovulatory phase compared to the other two phases [10,11,12], other studies have found no impact of MCP on tendon biomechanical properties [13,14], tendon fibril characteristics, or collagen cross-linking (8). A recent scoping review also describes that not only are menstrual cycle peaks of relaxin associated with matrix metalloproteinases (MMPs) activation and, hence, local collagen and gelatine degradation, but also, women have relaxin receptors in multiple joints, such that high relaxin levels correlate with greater joint laxity and a higher incidence of musculoskeletal injuries [15]. The inconsistency in findings between studies might result from varying measurement methods (e.g., validity of criterion outcome such as referring to tendon stiffness when joint laxity was assessed, measurement of serum vs. saliva hormone levels) and the timing of measurements (i.e., relative to menstrual cycle phase duration, which phases of the menstrual cycle have been contrasted, or whether sufficient delay has been allowed for any measurable hormonal impact). 

Previous authors Eiling et al. [14] found considerable change in muscle stiffness (i.e., muscle resistance to changes in muscle length) across the 28-day menstrual cycle, which might cause a difference in strength and jump height performance. Indeed, they reported greater knee laxity in the ovulation phase than in the follicular and luteal phases and linked this effect to high oestradiol and low progesterone in the ovulation phase. It is also important to note that pain perception may also alter across the MCPs due to oestrogen’s influence on sensory processes, with a greater pain sensitivity in the menstrual and pre-menstrual phases than in the mid-menstrual and ovulatory phases [16,17]. The modulation of pain plays a role in flexibility training since stretch tolerance affects physical performance and the amount of load tolerated during the physical procedure of stretching [18]. Studies have evaluated variables that affect flexibility performance, such as muscle–tendon unit range of motion (ROM) [19], tendon laxity [20], pain tolerance, and muscle–tendon unit stiffness [21]. However, no study has examined flexibility modification in a multi-factorial approach across the MCPs, especially in terms of the modulation of flexibility via muscle structure and/or function. Understanding any multiway interaction between these parameters (i.e., hormonal levels vs. flexibility vs. muscle-tendon unit structural characteristics vs. sensory factors) is essential for populations for whom flexibility is a crucial skill, such as for dance. 

Therefore, current study aimed to evaluate the effect of different MCP time points and/or hormonal changes in dancers on flexibility modulation by assessing muscle structure and function in both legs. Given the current focus on lower limb function, the semitendinosus (ST) was selected owing to its involvement in flexibility and jump performance. The hypothesis was that the muscle–tendon unit stiffness and pain tolerance would decrease with elevated levels of oestradiol (e.g., during the ovulation phase), leading to an increase in the ROM, whilst stiffness would increase with high levels of progesterone (e.g., during the luteal phase), thus improving jump performance.

## 2. Materials and Methods

Ethical approval was granted by Manchester Metropolitan University and performed according to the latest version of the Declaration of Helsinki [22], approval code (22.12.15 (ii). Participants provided written informed consent.

### 2.1. Participants

Eleven women (mean [SD]: age 23.5 [2.9] years, body mass 67.7 [15.6] kg, height 1.63 [0.05] m) were included in the study. Participants were contemporary dance students with an average of 10.5 [1.73] hours of dance practice and 6.12 [2.36] hours of other physical activity per week. The Physical Activity Readiness Questionnaire (ParQ) was used to uncover any potential health risks associated with exercise according to the ACSM Standards and Guidelines for Health and Fitness Facilities [23]. Inclusion criteria comprised the absence of injuries in the lower back and lower limbs in the last month or previous injuries that the research protocols could aggravate and the absence of pharmaceutical contraception for the last six months. Exclusion criteria were no formal dance background and recently (within last 6 months) or currently taking any medication likely to affect physical performance abilities.

### 2.2. Timing and Order of Procedures

Laboratory-based assessments occurred in four separate sessions: familiarisation with all procedures, booked at each participant’s first convenience; and three months later, following three months of body temperature, urinary ovulation assessment, and menstruation diaries, three further sessions were booked to coincide with mid-luteal, ovulation (within 1 day), and mid-follicular phases of one menstrual cycle. During familiarisation, participants trained for the flexibility and jump tests. Participants attended the phlebotomy laboratory at university during each test session in the morning after 12 h overnight fast. They were asked to drink 500 mL of water approximately two hours before data collection to standardise hydration levels (according to the ACSM recommendations). Serum and saliva samples were used to analyse hormone concentration. A trained phlebotomist collected blood from one of the antecubital fossa veins, and the participant allowed saliva to freely flow into a collection tube. Following the phlebotomy/saliva sampling procedures, participants had breakfast consisting of (naturally caffeine-free) fruit tea, water, two slices of whole grain bread with butter or jam, yoghurt, and fruit (approximately 250 kcal). Anthropometry measurements were performed, before the participant laid supine on a physiotherapy bed for the ultrasound recordings of the Semitendinosus (ST). The participant performed the jump pre-test consisting of three maximal counter movement jumps (CMJ), immediately followed by three maximal squat jumps (SJ). No warm-ups before the jumps were performed. Then, participants were positioned on the flexibility equipment test. Finally, participants undertook the pain mixed-method assessment. See below for details.

### 2.3. A priori Identification of Menstrual Cycle Phases, Length, and Regularity

A paper-based menstrual cycle calendar and a digital basal thermometer (Geratherm, Geratherm Medical, Geschwenda, Germany) with an accuracy of ±0.10 °C were given to participants to continuously track their menstrual cycle for three full months before the tests. The basal temperature was measured, to two decimal places, daily after waking up, and the days of menstruation were noted down. In addition, a urinary ovulation kit (One Step Ultra Early Pregnancy Tests at 10 mIU/mL, One + Step^®^, Shanghai International Holding Corp, Hamburg, Germany) was used to confirm ovulation. The kit consisted of strips of colorimetric enzyme immunoassays of urinary LH used daily starting five days before the predicted ovulation (inferred from the preceding days of paper-based temperature records). This was expected to exhibit an LH surge, suggestive of ovulation within 14–26 h [24]. Meanwhile, the luteal phase testing was at the mid-point between the ovulation measurement day and the next expected first day of menstruation with associated continued elevated temperature, and the follicular testing was immediately after the last day of menstruation. A 2-day window for testing was allowed for laboratory measurements at each phase to fit with participants’ availability.

### 2.4. Biochemical Analyses

Whole blood analysis of fasting plasma glucose was performed using an Accutrend Plus (Roche Diagnostics Limited, Welwyn Garden City, UK) monitoring device and Accutrend test strips (Roche Diagnostics Limited, Welwyn Garden City, UK) to ensure participants were in a fasted state (Lin’s coefficient: glucose = 0.958 [25]). Commercially available enzyme-linked immunosorbent assay (ELISA) kits were used to determine serum and saliva hormone concentrations, including oestrogen, progesterone, and relaxin. Serum coefficients of variation (CVs) were as follows: oestrogen 8.4–9.2% and relaxin 2.1–3.6% (R&D Systems, Bio-techne, Minneapolis, MN, USA); progesterone < 10% (Abbexa, Cambridge, UK). Saliva CVs were as follows: oestrogen 2.4–8.3% and progesterone 3.5–8% (Demeditec Diagnostics, Kiel, Germany).

### 2.5. Anthropometry and In Vivo Body Composition

Body height (to the closest 0.1 cm) and body mass (wearing light clothing, to the nearest 0.1 kg) were measured with participants standing in an orthostatic position using a wall-mounted stadiometer (Holtain Ltd., Crymych, UK) and a digital body mass scale (Sseca GmbH & Co. KG., Hamburg, Germany), respectively. This allowed the computation of body mass index (BMI). A bioelectrical impedance analysis (BIA) (BodyStat 500, Bodystat Ltd., Isle of Man, UK) was used to estimate relative body fat, lean mass, and basal metabolic rate. Calf, thigh, hip, and waist circumferences were assessed using a standard tape measure.

### 2.6. Muscle Architecture

ST ultrasound assessments (MyLabTM Gamma; Esaote, Reading, Berks, UK) were conducted with a scanning frequency of 7.50 MHz, in brightness mode (B mode) with a depth of penetration of 49.3 mm and a focus of 27.0–31.0. Live streaming of all assessments was captured by an HP computer running video capture software (Premier 6.0, Adobe Systems, San Jose, CA, USA) through an analogue-to-digital converter (Pinnacle, Corel Inc., Ottawa, Ont., Canada). Structural measures of the ST were taken at 50% length from the femur’s head to the lateral epicondyle and mid-width of the thigh. The medial and lateral boundaries of the muscle and its cross-sectional area (CSA) scans (Figure 1A) were identified in the transverse plane. Muscle thickness measurements (Figure 1B) (distance between the superficial and deep aponeurosis) were measured in the sagittal plane alongside the fat thickness and total thickness (from the subcutaneous adipose tissue–muscle interface to the muscle–bone interface). 

Accuracy of muscle thickness as described above has been previously established [26,27]. Ultrasound scans were recorded and digitised on an HP Windows laptop and analysed offline with digitising software Version 1.1.20123.0 (Dartfish for video capture, Gimp (Adobe, Maidenhead, UK) for digital image manipulation and ImageJ (National Institutes of Health and the Laboratory for Optical and Computational Instrumentation (LOCI), LOCI- Laboratory for Optical and Computational Instrumentation, The University of Wisconsin, Madison, WI, USA) for digital image measurement). 

### 2.7. Jump Performance Assessment

Two synchronised force platforms (AMTI, Watertown, MA, USA) mounted side by side with an acquisition frequency of 1000 Hz [28] and the software 2.6 Nexus Motion Capture (Oxford Metrics, LA, USA) were used to acquire jump ground reaction force. Three counter movement jumps (CMJ) followed by three squat jumps (SJ) were completed with a 20-s interval between trials. The highest CMJ and SJ were recorded as a measure of the participant’s jump performance. Peak force and impulse were combined from each of the force plates and used to obtain take-off velocity and jump height using the impulse–momentum relationship and equations of uniform acceleration, respectively. 

### 2.8. Semitendinosus Flexibility

Participants were positioned on a purpose-built flexibility machine designed to measure passive torque, passive ROM, and first sensation of stretch (FSS) in the right and left lower limb separately (Figure 2). Participants were positioned, supine, on the machine with their greater trochanter aligned with the lever rotation axis and their ankle supported 2 cm proximal to the lateral malleolus. The machine’s load cell (located under this ankle support) measured the ST resistance force whilst being stretched. Whilst in the supine position, the hip was considered to have 0° of flexion and the knee was maintained at 0° of flexion while the hip was being flexed. With straps on the ankle, the distal third of the thigh and anterior superior iliac spines were used to maintain the supine position and extended knee. The contralateral limb was strapped to the table, and cushions underneath the lower back and neck were used both for comfort and to minimise any compensatory movements. 

The participant interacted with the machine in two ways: (1) by pressing primary control buttons to rotate the lever arm to which the lower limb was attached, thereby stretching the ST during (passive hip) flexion; (2) by pressing a secondary control button at the first sensation of stretch, i.e., tension in the ST. For gravity correction, used to adjust the torque values, participants lay supine, and the mass of their lower limb was measured with the hip at 0° hip flexion. The potentiometer, the load cell, and the FSS control button were connected to an analogue/digital converter (NI USB-6008 National Instruments, Austin, TX, USA), connected to a desktop computer (Porgété Z30, Toshiba, Hammfelddamm, Neuss, Germany). The Dasylab program 11.0 (Dasytec Daten System Technik GmbH, Ludwigsburg, Germany) was used for data acquisition and analysis. 

Flexibility tests consisted of a series of six passive stretches at 5 °/s, controlled by the participant, until reaching the maximal hip flexion ROM that was tolerated by the participant (ROMMax). At this point, the acquired passive torque value was defined as torqueMax. Participants would press the second control button when they perceived the first sensation of stretch (FSS) by feeling the hamstrings’ tension. This way, ROM and torque respective values were noted as FSSROM and FSStorque. The muscle–tendon unit passive stiffness (SMTU) was calculated as the torque variation divided by the ROM variation [29] in the third portion of the length–tension curve obtained during the stretch to guarantee no muscle activation. Elastic potential energy was calculated as the area under this curve. Participants were blindfolded to avoid any visual stimuli interfering with their stretch tolerance. The reliability for the six trials of ROMMax, torqueMax, FSSROM, and FSStorque were assessed via intraclass correlation coefficients and ranged from good to excellent for all the variables (>0.67 and <0.85) [30].

### 2.9. Muscle Electromyographical Activity

EMGs from the ST and rectus femoris (RF) were measured during both the flexibility and jump tests using surface electromyography (Trigno, Delsys, Natick, Massachusetts, USA) using a frequency of acquisition of 1000 Hz and an amplification of ×1000. ST electrodes were positioned at the medial point of the ischial tuberosity and the medial epicondyle of the femur [31,32] and the RF electrodes at the medial point of the RF tendon and the patella [33]. Isometric muscle contractions were performed to check the signal quality during hip flexion and hip extension. Data processing began with rectification following the removal of any offsets and then conversion to root mean square (RMS) with a window of 0.1 s and overlap of 0.08 s. Subsequently, the muscle activation value relative to resting muscle activation (%RVC) was used as a normalisation protocol for dynamic efforts (i.e., in the flexibility and jump tests) to allow for comparison between participants. The resting value was chosen at the point when the participant was prepared to perform the jump tests but before any movement had begun. For the flexibility analyses, any EMG signal exceeding the resting baseline value plus twice the standard deviation was excluded [20,21]. Establishment of this threshold of low EMG activity was used to confirm the stretch’s passive nature in the analysed signals.

### 2.10. Pain Mixed-Method Assessment

Finally, participants undertook the pain mixed-method assessment where they were randomly assigned to the Ice Water Test (IWT) followed by completing questionnaires, or vice versa, to minimise any order effect. The IWT was performed to characterise participants’ sensitivity to pain and involved two water containers sufficiently deep to allow the dominant forearm’s immersion up to the elbow, one with cold water and the other at body temperature. A general purpose liquid-in-glass thermometer ranging from −10 to 110 °C, 50 mm immersion (B60300-0000, H-B Instrument, Loughborough, Leicestershire UK), was immersed in each container to ensure the temperature remained at 35–39° Celsius (body temperature) or −3–0° Celsius (cold sensation). Each participant’s dominant forearm was immersed in the body temperature container for 120 s to standardise initial conditions before the IWT. Participants were instructed to hold for as long as possible with the arm under the water and take their arms out of the water whenever they felt they could no longer tolerate the cold. A digital chronometer recorded the time of withdrawal or the cut-off threshold at 120 s. 

Qualitative (affective and sensory) aspects of the possible pain experienced during cold and warm water immersion were assessed using the Visual Analogue Scale (VAS). The scale was shown to participants at 15 s intervals while their arm remained in the cold water to obtain the representative level of discomfort from zero (no pain) to ten (the maximal pain ever felt). The SEFIP (Self-Estimated Functional Inability Because of Pain questionnaire) was designed for dancers [34,35] in injury and pain research utilizing a body map to localise pain, whereby 16 body areas are rated on a 5-point Lickert scale. The short form 20 version of the PASS (Pain Anxiety Symptoms Scale) [36,37] assessed four subscales referred to as dimensions in pain research: cognitive anxiety, escape and avoidance, fear, and physiological anxiety. This questionnaire produces two outcomes (TotalPASS, i.e., the total score, and PASSPhysio, i.e., physiological anxiety score).

### 2.11. Data Reduction

To identify if any asymmetry level between limbs would vary across the menstrual cycle, delta [D − nD)/D] was calculated (where D is the dominant limb and nD the non-dominant limb, with the dominance determined as the limb with the largest range of motion—ROM). Not only is asymmetry an interesting physical function marker of itself, but soft tissue asymmetry has also been shown to vary with changes in luteinising hormone [38]. A normalisation of all dependent variables and hormone concentrations values was made by expressing data relative to values at the follicular phase. This is because both progesterone and oestrogen levels are expected to be low; therefore, changes could be highlighted in the following phases. In addition, differences in these normalised values were statistically assessed (see below) between each menstrual cycle phase, and, where significant, changes were then correlated against relative changes in hormones.

### 2.12. Outcome Variables

Table 1 summarises the assessed variables in each phase of the three phases of the menstrual cycle for the dominant and non-dominant limbs.

### 2.13. Statistical Analyses

SPSS Statistics (v24 International Business Machines Corporation, New York, NY, USA) was used for statistical analyses. Levene and Shapiro–Wilk statistical tests were performed to test homogeneity of variance and normality of the data, respectively. A comparison between the flexible (dominant limb—D) and least flexible (non-dominant limb—nD) lower limb (hereafter referred to as leg dominance) for all the dependent variables across the MCPs (ovulatory, follicular and luteal) was performed using a repeated measures ANOVA with six levels of IVs (2 limbs, each at 3 time points) (when parametric assumptions were met) or the Friedman test (when assumptions were not met). Pair-wise post hoc t-tests or Wilcoxon comparisons were performed to highlight which pairs were the basis for the main effect. 

A second analysis using the repeated measures ANOVA with three factors (when parametric) or the Friedman test (when non-parametric) was performed to compare the relative change (i.e., delta) between limbs [(D − nD)/D] for each variable between the MCPs. Relative change in all dependent variables and relative change in hormone concentrations were correlated using values from the follicular phase as a baseline. Finally, co-variance analyses (ANCOVA) were performed to evaluate the hormonal influence on the dependent variables and, hence, correct for covariates where appropriate. The statistical significance adopted was α ≤ 0.05, and adequate study power was determined when β ≥ 0.8 (and effect size was notable when pε^2^ ≥ 0.2. Note that pε^2^ was only computed when study power was adequate).

## 3. Results

### 3.1. Descriptive Analysis

The characteristics of the dancers during three time points in the menstrual cycle phases are shown in Table 2. Notably, the average duration of the menstrual cycle over the three months of monitoring was 32 ± 7 days (range of 23 to 45 days), with measurements on day 5 ± 2 for follicular, day 16 ± 4 for ovulation, and day 24 ± 6 for luteal. Most variables presented homogeneity of variance with the exception of FSStorque (*p* = 0.037) and peak force (*p* = 0.018) for the CMJ in the non-dominant limb, total peak force (*p* = 0.005) for the SJ, upper back (*p* = 0.015), back thighs (*p* = 0.036), shoulders (0.001), and ankles/feet (*p* = 0.004).

### 3.2. Hormonal Variation between Three Menstrual Cycle Phases

Despite the trends showing oestrogen to be the greatest during the ovulatory phase, it was remarkable that, due to large inter-individual variance, no significant differences between the phases were found for oestrogen, progesterone, and relaxin levels across the menstrual cycle phases (Figure 3).

### 3.3. Structural, Functional, and Pain Sensation Characteristics across Three Menstrual Cycle Phases and Limb Comparisons

Only two of the outcome variables, TotalPASS (F3.515 *p* = 0.049; η2p = 0.260; β = 0.59) and PASSPhysio (F7.219 *p* = 0.009; η2p = 0.419; β = 0.82), significantly differed when structural, functional, and pain sensation characteristics were compared across menstrual cycle phases. TotalPASS was found to be greater during the ovulation phase when compared to the luteal phase (*p* = 0.009). No differences, however, were found either between luteal and follicular (*p* = 0.416) or ovulatory and follicular (*p* = 0.060) phases. The PASSPhysio score was found to be significantly greater in the ovulatory phase compared to the luteal (*p* = 0.023) and follicular phases (*p* = 0.001). No difference was found between follicular and luteal phases (*p* = 0.250). In addition, only ROMMax showed a difference between limbs (F4.157 *p* = 0.019; η2p = 0.294; β = 0.76). 

The ROMMax in the dominant limb was significantly higher than the non-dominant limb in all the menstrual cycle phases (follicular *p* = 0.004, ovulatory *p* = 0.001, and luteal *p* = 0.001), although no significant differences were found in this variable across the phases (Table 2).

### 3.4. Relative Changes and Correlations between Change in Outcome Variables and Change in Hormone Levels

A secondary analysis was performed to identify if any existing relative asymmetry between limbs (i.e., delta [(D − nD)/D]) would vary across the menstrual cycle. No relative limb differences (delta) were found between the phases.

Figure 2 shows whether each hormone’s relative change (i.e., DELTA) in the luteal/follicular phase differed from its DELTA in the ovulatory/follicular phase. Relaxin was the only hormone which presented a significant difference in the variation between ovulation/follicular and luteal/follicular phases.

Table 3 shows only the statistically significant correlations between hormones and structural, functional, and pain sensation characteristics. It is, thus, remarkable that oestrogen change was associated with a change in 11 musculoskeletal outcome variables, progesterone change was associated with a change in 6 muscle–tendon characteristic outcome variables, and relaxin change was associated with a change in 14 muscle–tendon characteristic outcome variables. 

## 4. Discussion

The current study aimed to evaluate any modulatory effect of different time points and/or hormonal profile changes across the MCP on flexibility, muscle structure, and function in dancers. It was hypothesised that in the ovulatory phase, the ROMMax would be increased, and in the luteal, the ROMMax would be decreased due to the variation in hormonal concentrations of oestrogen, progesterone, and/or relaxin in each phase. Thus, the structural and functional characteristics of the MTU were expected to be affected. Previous literature found significantly higher oestradiol levels during the post-ovulatory and mid-luteal phases than the menses, and progesterone levels were significantly lower during the menses and post-ovulatory phases than the mid-luteal phase [39]. In contrast, in the present study, there was no significant group change in menstrual hormones through the menstrual cycle phases; this could be due to (a) large inter-individual variability or (b) menstrual cycle irregularities. Previous research found irregularities in dancers’ menstrual cycle [40,41,42,43,44,45]. In addition, changes in habitual exercise levels [46] and even the emotional status of an individual [47,48] may add to both inter- and intra-individual variability, making it challenging to reach the targeted assessment phase. Unless days are counted in retrospect, it is difficult to predict the exact day of ovulation [4]; therefore, any variation in either habitual physical activity or mental status might have affected our accuracy in testing at the true hormonal peak and/or accounting for any delay in the physiological effects of hormonal variations. The range of different research methods [49], such as timing and number of phases tested [50,51,52,53,54] and heterogeneity in oestrogen levels [55], helps to obscure possible MCP hormonal effects on exercise performance, since their potential effect is most likely to be found during the phases with no hormonal dysregulation, i.e., significantly different hormone levels across the menstrual cycle [4].

### 4.1. Accuracy of Targeting of Hormonal Peaks within the MCP

Some, but not all, ovulating women have a body basal temperature increase of approximately 0.3 °C after ovulation, sustained throughout the luteal phase [56,57,58,59]. Although in the current study, the temperature increases after ovulation averaged to 1.60 ± 0.16 °C across the MCPs, there was high variability in this outcome measure. In a 30-year analysis of human MCP temporal characteristics, the average 28-day menstrual cycle was unsupported [60], varying substantially [60,61,62]. Each woman has a central trend and variation, which changes with age [60,63]. Inconsistencies from one menstrual cycle to another do not necessarily reflect alterations in the bleeding pattern [64]; therefore, participants with menstrual abnormalities could be considered normal without in-depth hormonal analysis [65].

The ovulation kit aims to detect the urine-increased LH level just before ovulation with 99% accuracy and a correlation between 68–84% to the gold standard methods of predicting ovulation [66,67]. However, the urine strips’ detected ovulation does not always coincide with the peak in the basal body temperature [68]. Potential sources of error include improper test kit performance, differing kit sensitivities, individual test kit variation, variation in the amplitude and duration of LH surges, and variation in the interpretation of the test window colour [24,66,67]. Notwithstanding potential shortcomings in oestrogen testing, the longitudinal pre-test monitoring within the current study minimised any issues with targeting the planned menstrual cycle phases. It does not, however, detract from the possibility of hormonal dysregulation and/or the possibility of a spontaneously sporadic anovulatory cycle in our study sample [69], despite their regular menstruations and urine-based ovulation marker tracked over three months.

### 4.2. Inferences from Grouped Raw Data

No differences were observed for the ST CSA, fat thickness, ST thickness, total body lean mass, thigh width, and thigh length across the MCPs in the current study. It is interesting to note that the ST CSA has been shown to decrease with long term reductions in oestrogen due to menopause [70]. Postmenopausal women tend to be susceptible to many deleterious changes in musculoskeletal characteristics, unless undergoing hormone replacement therapy [71], suggesting that oestrogen may have a muscle-strengthening action [71,72]. Phillips et al. [71] found muscle CSA to vary significantly between MCPs. However, the CSA was measured anthropometrically using callipers, while the ST CSA in the current study was acquired via ultrasound imaging. Despite the higher precision with ultrasound imaging compared to anthropometry, no differences were found in the present study across the MCP. Previous authors [73,74] also did not find differences in weight, per cent body fat, the sum of skinfolds, haemoglobin concentration, haematocrit, maximum heart rate, maximum minute ventilation, maximum respiratory exchange ratio, anaerobic performance, endurance time to fatigue (at 90% of VO2max), or isokinetic strength of knee flexion and extension in between the luteal and follicular phases, corroborating the current study. In contrast, a study with daily body weight measurements in 28 young women found the highest body weight in the late luteal phase and in the first days of menstruation, followed by abrupt weight loss. A short increase peak in body weight after ovulation was also found by other authors [75]. The increase in body weight across the menstrual cycle might be related to fluid retention. A study examining daily self-reported “bloating” over one year found peak retention on the first day of menstrual flow. However, neither oestradiol nor progesterone levels were significantly associated with this bloating [76].

The absence of anthropometric and MTU structural differences in the present study were consistent with the finding that there was no menstrual variation in the vertical jump and flexibility variables. Bell et al. [77] found that hamstring MTU stiffness did not change across the menstrual cycle, corroborating other authors’ findings [78,79]. Muscle strength might be related to oestrogen peak [80] with an increased incidence of anterior cruciate ligament injuries [81] due to the increased joint laxity [11,20]. Notwithstanding this, no other strength-related variable differed between the phases. Phillips et al. [71] measured muscle strength throughout the menstrual cycle, detecting ovulation by urine luteinising hormone measurements or change in basal body temperature, and found a significant increase in strength during the follicular phase when oestrogen levels rose and a significant decrease in strength around ovulation.

No correlation between plasma oestrogen and muscle force was found; oestrogen action on the muscle might take hours or days to occur. In previous work, neuromuscular and biomechanical characteristics were not influenced by oestradiol and progesterone fluctuations [39]. MCP variations and the use of oral contraception did not affect knee or hip joint loading during jumping and landing tasks [82]. Confounding variables require the use of caution when comparing studies; for example, progesterone concentration is higher in the morning [83], and post-exercise status may increase oestrogen and progesterone serum concentrations [84,85].

The ROMMax was statistically different between the limbs across all phases, with no difference between the phases. Delta comparison [(D − nD)/D] confirmed that the asymmetry level does not vary across the MCP. If the circulating hormonal levels variation affects ROM, both limbs are equally affected, with a similar result for the other variables. The total PASS and PASS Physiological Anxiety subscale in the ovulatory phase were more significant than the luteal, with no difference between the other phases. The ovulation higher score suggests greater fearful appraisals of pain [86]. Although the PASS Physiological Anxiety subscale is related to the bodily reaction when experiencing or anticipating pain and was shown to be higher at ovulation, corroborating the pain research across the menstrual cycle, neither the torqueMax nor the FSStorque, variables associated with the stretch tolerance, were shown to be different between the phases or the limbs. Stretch pain is associated with stretch stimulations in soft tissue; controlling it helps ROM increase [87]. Each MCP may be related to a variety of behavioural outcomes, such as the perception of attention, memory, and pain [88]. However, none of those factors appeared to affect flexibility across the phases.

### 4.3. Inferences from Individually Normalised Data

A ratio of all dependent variables and hormone concentrations was performed using values from the follicular phase as the baseline due to low levels of progesterone and oestrogen; therefore, changes could be highlighted in the following phases. Also, between MCPs, differences in these relative changes were statistically assessed. When those changes were significant, they were correlated against relative changes in hormones within each female. It was, thus, highly informative to find a substantial number of significant associations, potentially indicative of a causal effect of hormones on these outcome variables. Although no group difference in the hormonal concentration was found across the MCPs, Δ oestrogen luteal/follicular was negatively correlated with Δ muscle length, Δ muscle CSA, Δ fat thickness, and Δ muscle activity (EMG) during the CMJ, while Δ oestrogen ovulatory/follicular was positively correlated with Δ ST thickness, Δ FSStorque, and ΔCMJ EMGRF. Δ progesterone showed the opposite behaviour for similar variables, being positively correlated when oestrogen was negatively correlated and vice versa. These results, the direction of the correlations, and the dependent variables that the hormones are correlated to corroborate findings from the literature suggesting an MTU-loosening effect of oestrogen and a tightening effect of progesterone. Although relaxin is more prominent after pregnancy, its variation seems related to the tissue’s laxity, corroborating previous literature [89]. Notably, relaxin was the only hormone which presented a significant difference in the variation between ovulation/follicular and luteal/follicular phases (see Figure 3). It is important to note that this study was underpowered and should be repeated in a larger sample size.

### 4.4. Study Limitations and Recommendations for Future Work

Whilst the current study was rigorously conducted, it nonetheless has some limitations. First, participants self-reported their LH surge as the urine tests were unsupervised. It is plausible that some may have simply misreported this data point, hence impacting the scheduling of laboratory tests. It is recommended that future studies have an element of supervision for the daily temperature and other tests so as to minimize this reliance on participants’ reports. Second, whilst the sample size proved powered for a number of our outcome measures, it was still modest; hence, it inhibited sub-grouping where only those with hormonal changes could be segregated for further analyses. Thus, future studies in larger samples are required.

## 5. Conclusions

No differences in the grouped data in circulating female hormones were found across the MCP, highlighting the heterogeneity of menstrual cycle phases in dancers. There is also the possibility of hormonal dysregulation in our study sample, despite their regular menstruations and urine-based ovulation-marker tracked over three months. There were also no differences in limb asymmetries across the MCP. However, the relative individual changes in hormonal levels were associated with most of the relative changes in the key outcome measures, including the structural and functional characteristics of the MTU. Thus, a negative correlation between progesterone and flexibility and a negative correlation between oestrogen and jump variables follow the suggested role of these hormones. Furthermore, oestrogen and relaxin were associated with increased MTU compliance, whilst progesterone was associated with increased muscle stiffness. Relaxin was correlated with even more outcome variables. It was the only hormone which presented a significant association with the variation between ovulation/follicular and luteal/follicular phases, highlighting its essential role in MTU laxity beyond pregnancy time.

## Figures and Tables

**Figure 1 jfmk-09-00038-f001:**
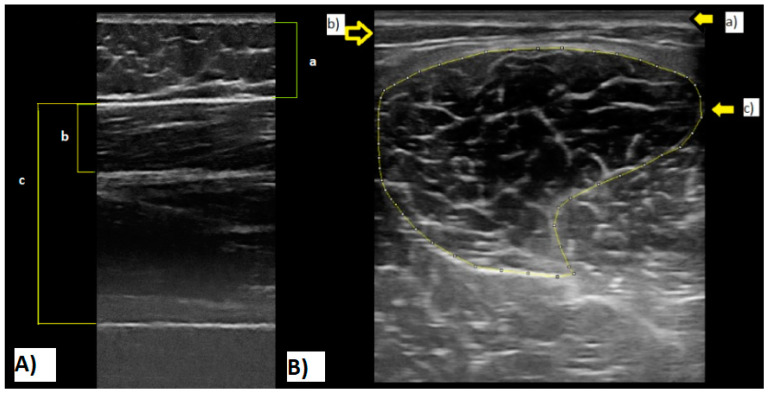
(**A**): Semitendinosus ultrasound cross-sectional area image. (a) Skin; (b) subcutaneous fat; (c) muscle aponeuroses. (**B**): Ultrasound image. (a) Fat thickness; (b) semitendinosus thickness; (c) lean total thickness. Note: All images were recorded during tests of the current study.

**Figure 2 jfmk-09-00038-f002:**
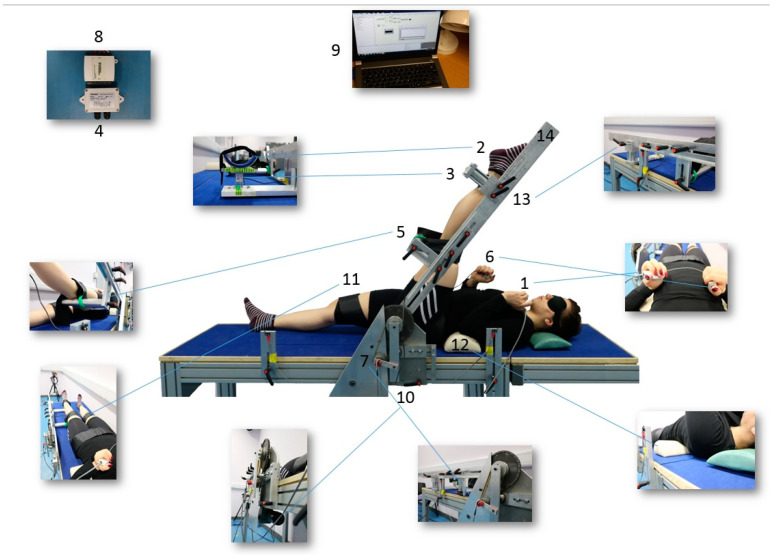
1. Push button to control the ascending and descending movements of the lever. 2. The ankle support designed in a “U” shape to minimise hip external rotation. 3. Load cell (CS 15 V, Líder Balança, Araçatuba, SP, Brazil) to measure the MTU’s resistance force against stretch. 4. Amplifier (Strain Gauge Transducer SMOWO, RW-ST01, Shanghai Tianhe Automation Instrumentation Co., Shanghai, China). 5. Support for the thigh to avoid hyperextension of the knee. 6. Controller to signal the FSS: a tension in the hamstrings. 7. Potentiometer (TT Electronics ABW1 5K +/− 10% Rapid Electronics part no 51-7053, Abercynon, United Kingdom TT) to record the ROM. 8. Analogical/digital converter (NI USB-6008 National Instruments). 9. Computer: Dasylab program 11.0 (Dasytec Daten System Technik GmbH, Ludwigsburg, Germany). 10. Motor (Parvalux motor and right angle gearbox model BH11 8PU PM3d LWS63690/01J, Parvalux, Bournemouth, United Kingdom). 11. Straps to fix the limb. 12. Cushions for the neck and lumbar areas. 13. Adjustable sections according to participant’s limb length. 14. Lever. (Photos: Bárbara Pessali-Marques).

**Figure 3 jfmk-09-00038-f003:**
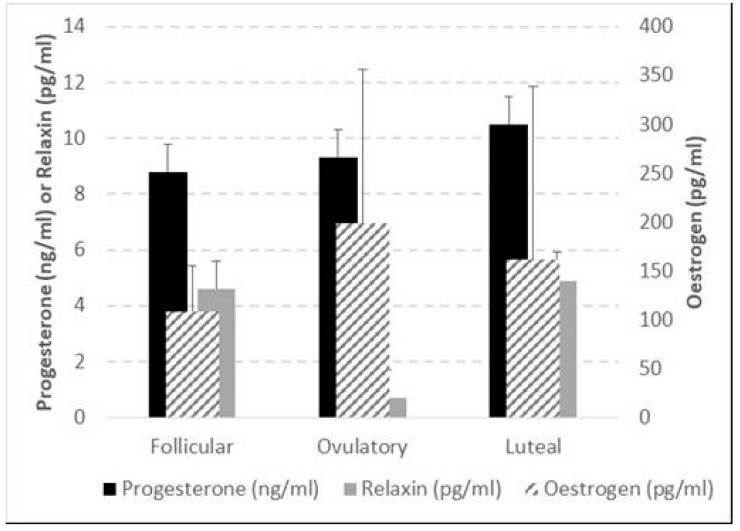
Average and standard deviation of oestrogen, progesterone, and relaxin at follicular, ovulatory, and luteal phases of the menstrual cycle. The bracket indicates the only significant difference, i.e., the Wilcoxon-based comparison of the relaxin changes in luteal—ovulatory vs. luteal—follicular phases of the menstrual cycle (*p* = 0.001). For progesterone, the lowest *p* = 0.097; whilst for oestrogen the lowest *p* = 0.271.

**Table 1 jfmk-09-00038-t001:** Outcome variables assessed in each menstrual cycle phase for dominant and non-dominant limb.

Flexibility	Vertical Jump	Pain Mix Method	EMG	Ultrasonography	Serum, Saliva, and Whole Blood
ROM_Max_ Torque_Max_ FSS_ROM_ FSS_torque_ S_MTU_	Jump height Impulse GRF Force_peak_ v_take-off_	SEFIP PASS VAS Ice Water Test	EMG_ST_ EMG_RF_ during CMJ and SJ and flexibility tests	CSA Length Width Fat thickness Lean thickness Semitendinosus thickness	Oestrogen Progesterone Relaxin (serum) Glucose

ROM: range of motion, Max: maximal, FSS: first sensation of stretch, S: stiffness, MTU: muscle–tendon unit, GRF: ground reaction force, V: velocity, SEFIP: Self-Estimated Functional Inability because of Pain, PASS: Pain Anxiety Symptom Scale, VAS: visual analogue scale, EMG: electromyography, RF: rectus femoris, ST: semitendinosus, CMJ: countermovement jump, SJ: squat jump, CSA: cross-sectional area.

**Table 2 jfmk-09-00038-t002:** Descriptive analysis of the dancers across three menstrual cycle phases (average ± standard deviation). Note that there are no statistically significant differences in raw data.

	Follicular		Ovulatory		Luteal	
Age (years)	23.5 ± 2.94					
Height (m)	1.63 ± 0.05					
Body mass (kg)	67.5 ± 16.0		67.6 ± 15.6		67.8 ± 16.0	
Fat%	25.35 ± 4.53		30.3 ± 6.8		30.81 ± 6.03	
Lean%	69.50 ± 6.94		69.7 ± 6.8		69.18 ± 6.03	
Estimated basal metabolic rate (j)	6487.20 ± 636.34		6460.7 ± 588.1		6449.00 ± 696.53	
Body mass index (Kg/m^2^)	25.29 ± 4.62		25.4 ± 4.5		25.35 ± 4.53	
Fasted glucose (mmol/L)	5.80 ± 2.97		6.06 ± 3.61		5.04 ± 1.16	
Oestrogen (pg/mL)	108.7 ± 46.4		199.28 ± 157.4		161.4 ± 177.3	
Progesterone (ng/mL)	8.8 ± 2.6		9.3 ± 3.0		10.5 ± 2.1	
Relaxin (pg/mL)	4.6 ± 1.3		0.7 ± 0.4		4.9 ± 6.6	
Hips circumference (cm)	98.2 ± 10.6		98.8 ± 10.1		100.9 ± 6.1	
Waist circumference (cm)	79.7 ± 16.9		78.8 ± 16.6		77.5 ± 15.3	
	Dominant	Non-dominant	Dominant	Non-dominant	Dominant	Non-dominant
Calf limb circumference (cm)	36.72 ± 4.06	36.86 ± 3.67	37.11 ± 6.40	37.11 ± 6.39	36.22 ± 7.84	36.05 ± 7.84
Thigh circumference (cm)	53.31 ± 3.30	53.37 ± 3.75	51.72 ± 6.40	51.54 ± 6.61	51.31 ± 6.17	51.31 ± 5.96
ROM_Max_ (°)	135.7 ± 11.98	128.6 ± 13.9	135.6 ± 14.4	130.5 ± 12.8	136.9 ± 14.2	132.5 ± 39.7
Torque_Max_ (Nm)	121.3 ± 37.3	134.3 ± 40.0	129.9 ± 40.5	141.7 ± 36.9	134.7 ± 39.9	136.5 ± 39.7
FSS_ROM_ (°)	102.2 ± 15.8	94.9 ± 13.9	99.0 ± 10.5	96.7 ± 9.8	98.9 ± 9.5	91.6 ± 11.1
FSS_torque_ (Nm)	32.0 ± 13.3	49.2 ± 31.5	46.4 ± 21.6	55.3 ± 29.6	46.7 ± 29.4	34.5 ± 25.9
S_MTU_ (Nm/°)	2.9 ± 1.0	3.3 ± 0.9	3.1 ± 1.3	3.4 ± 0.9	3.2 ± 0.8	3.3 ± 1.3
Elastic potential energy (Nm°)	307.3 ± 105.3	302.2 ± 90.6	334.1 ± 122.6	342.6 ± 111.0	325.9 ± 93.7	341.9 ± 154.4
ST muscle length (cm)	39.2 ± 2.6	39.3 ± 2.5	39.3 ± 1.9	39.5 ± 2.0	39.6 ± 1.9	39.3 ± 1.7
ST muscle width (cm)	3.8 ± 0.5	3.7 ± 0.6	3.8 ± 0.8	3.82 ± 0.6	3.9 ± 0.9	4.1 ± 1.1
ST CSA (mm^2^)	5.2 ± 1.4	5.3 ± 1.2	5.6 ± 1.8	5.4 ± 2.0	5.4 ± 1.8	5.2 ± 1.8
ST fat thickness (mm)	1.3 ± 0.6	1.4 ± 0.6	1.4 ± 0.5	1.4 ± 0.5	1.13 ± 0.4	1.2 ± 0.5
ST muscle lean (mm)	5.2 ± 0.6	5.3 ± 0.8	5.4 ± 0.4	5.4 ± 0.3	5.3 ± 0.6	5.2 ± 0.8
CMJ impulse (Ns)	85.7 ± 32.9	64.0 ± 26.3	70.9 ± 23.8	76.9 ± 15.5	73.0 ± 15.4	74.5 ± 22.9
CMJ ground reaction force (N)	349.0 ± 97.3	356.1 ± 71.3	360.3 ± 95.6	350.4 ± 73.8	376.9 ± 95.7	332.5 ± 64.6
CMJ peak_force_ (N)	412.7 ± 107.7	364.0 ± 83.3	404.6 ± 73.3	385.8 ± 72.3	388.5 ± 64.1	402.4 ± 72.0
CMJ v_take-off_ (m/s)	2.10 ± 0.26		2.05 ± 0.25		2.05 ± 0.27	
CMJ height (m)	0.23 ± 0.06		0.22 ± 0.05		0.22 ± 0.06	
SJ impulse (Ns)	122.3 ± 52.3	106.3 ± 36.7	114.8 ± 40.16	122.6 ± 31.9	123.6 ± 68.0	121.1 ± 36.9
SJ ground reaction force (N)	363.1 ± 102.5	343.7 ± 62.5	357.8 ± 94.1	354.0 ± 74.6	372.2 ± 94.8	336.2 ± 65.0
SJ peak_force_ (N)	426.9 ± 123.8	386.6 ± 105.1	429.4 ± 127.6	402.5 ± 96.9	403.5 ± 76.8	393.3 ± 90.8
SJ v_take-off_ (m/s)	1.98 ± 0.26		1.99 ± 0.28		2.07 ± 0.25	
SJ height (m)	0.20 ± 0.05		0.21 ± 0.06		0.22 ± 0.05	

ROM: range of motion, Max: maximal, FSS: first sensation of stretch, S: stiffness, MTU: muscle-tendon unit, ST: Semitendinosus, CMJ: countermovement jump, SJ: squat jump, CSA: cross-sectional area.

**Table 3 jfmk-09-00038-t003:** Correlation between the outcome measures and hormonal changes. Pearson (when parametric) and Spearman (when non-parametric) significant correlations only are detailed. Data presented are P statistics.

		Oestrogen	Progesterone	Relaxin
Significant correlations Luteal/Follicular	Muscle length	*p* = 0.008 r = −0.560 **	*p* = 0.022 r = 0.587 *	n.s.
	Muscle CSA	*p* = 0.044 r = −0.413 *	n.s.	n.s.
	Fat thickness	*p* = 0.022 r = −0.480 *	*p* = 0.044 r = 0.513 *	*p* = 0.007 r = 0.683 **
	Lean	n.s.	n.s.	*p* = 0.015 r = 0.626 *
	CMJ EMG_RF_	*p* = 0.010 r = −0.611 *	n.s.	n.s.
	CMJ EMG_ST_	*p* = 0.001 r = −0.926 **	*p* = 0.006 r = 0.822 **	n.s.
	SJ EMG_RF_	n.s.	n.s.	*p* = 0.017 r = 0.790 *
	SJ EMG_ST_	n.s.	n.s.	*p* = 0.002 r = 0.911 **
Significant correlations Ovulatory/Follicular	Muscle length	*p* = 0.004 r = 0.599 **	n.s.	*p* = 0.049 r = 0.460 *
	Muscle CSA	n.s.	n.s.	*p* = 0.006 r = −0.646 **
	Fat thickness	n.s.	n.s.	*p* = 0.006 r = −0.647 **
	ST thickness	*p* = 0.001 r = 0.676 **	*p* = 0.010 r = −0.612 *	*p* = 0.001 r = −0.872 **
	Lean Muscle	n.s.	*p* = 0.045 r = −0.470 *	*p* = 0.001 r = −0.881 **
	FSS_torque_	*p* = 0.020 r = 0.463 *	n.s.	*p* = 0.028 r = −0.485 *
	S_MTU_	n.s.	n.s.	*p* = 0.001 r = −0.781 **
	Elastic potential Energy	n.s.	*p* = 0.034 r = −0.467 *	*p* = 0.021 r = −0.512 *
	CMJ Force_Peak_	n.s.	n.s.	*p* = 0.022 r = −0.509 *
	CMJ Total Force_peak_	*p* = 0.040 r = −0.578 *	n.s.	n.s.
	SJ Force_Peak_	*p* = 0.021 r = −0.459 *	n.s.	n.s.
	CMJ EMG_RF_	*p* = 0.021 r = 0.549 *	n.s.	n.s.
	SJ EMG_RF_	n.s.	n.s.	*p* = 0.007 r = 0.812 **

*p*: level of significance obtained, r: correlation, *: Correlation is significant at the 0.05 level (1-tailed). **: Correlation is significant at the 0.01 level (1-tailed). n.s.: not significantly different, CMJ: countermovement jump, SJ: squat jump, ST: semitendinosus, RF: rectus femoris, CSA: cross-sectional area, ROM: range of motion, Max: maximal, FSS: first sensation of stretch, SMTU: muscle–tendon unit stiffness, EMG: electromyography. Grey cells: Spearman’s correlation, white cells: Pearson’s correlation.

## Data Availability

Upon acceptance to be published, all data associated within this manuscript will be made available as either a ZIP folder to the publisher or a direct hyperlink to a digital storage resource within Manchester Metropolitan University.

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
