# Peer review of "Musculoskeletal Morphology and Joint Flexibility-Associated Functional Characteristics across Three Time Points during the Menstrual Cycle in Female Contemporary Dancers"

_jfmk, 2024, doi:10.3390/jfmk9010038_

Round 1
Reviewer 1 Report
Comments and Suggestions for Authors
Thank you for giving me the opportunity to review the manuscript entitled ‘Musculoskeletal morphology and joint flexibility-associated functional characteristics across three time points during the menstrual cycle in female contemporary dancers’. This manuscript aimed to assess the effect of menstrual cycle phase on muscle structure and function in 11 contemporary dancers. While this manuscript is well written, there are some major issues that should be addressed.
The procedures for determining menstrual cycle phase and timing of testing are not clear. There is mention of daily testing for urinary LH however the authors have only said that testing ‘should’ occur 5 days prior to predicted ovulation. Did this occur? And if so, when was testing scheduled for the ovulatory phase (for example, did this occur within 24 hours of the LH surge?). From the results, it appears that testing for the ovulatory phase occurred on day 14 for all participants however as the range of menstrual cycle length was from 23 to 45 days, it is highly likely that the ovulatory phase was missed in a number of participants. This would also help explain why there was no significant difference in hormone levels over the three phases. It is also not clear how the timing for testing of the luteal phase was determined. There is also an issue with the title and the aim of the study to assess over the phases of the menstrual cycle since the expected hormonal fluctuations to confirm menstrual cycle phase did not occur.
Since there were no differences in the group data, individual changes were assessed. Did the authors consider excluding the participants who did not experience the expected hormonal fluctuations (see https://pubmed.ncbi.nlm.nih.gov/31246715/#:~:text=Conclusion%3A%20To%20improve%20the%20quality,of%20serum%20estrogen%20and%20progesterone and https://pubmed.ncbi.nlm.nih.gov/33725341/ for recommendations)? It could be that the inclusion of participants who did not experience the expected fluctuations or who were not in the correct phase at the time of testing is masking group differences. It may be worthwhile looking to see how many participants would be included based on their hormone levels and re-running the group analysis.
Author Response
R1Q1: Thank you for giving me the opportunity to review the manuscript entitled ‘Musculoskeletal morphology and joint flexibility-associated functional characteristics across three time points during the menstrual cycle in female contemporary dancers’. This manuscript aimed to assess the effect of menstrual cycle phase on muscle structure and function in 11 contemporary dancers. While this manuscript is well written, there are some major issues that should be addressed.
R1R1: We thank our reviewer for taking the time to go through our manuscript. We have addressed each of their comments/queries on a point-by-point basis below.
R1Q2: The procedures for determining menstrual cycle phase and timing of testing are not clear. There is mention of daily testing for urinary LH however the authors have only said that testing ‘should’ occur 5 days prior to predicted ovulation. Did this occur? And if so, when was testing scheduled for the ovulatory phase (for example, did this occur within 24 hours of the LH surge?).
R1R2: The sentence was poorly phrased. Yes testing did occur daily from 5 days before expected ovulation. The instructions to participants was to contact us when said surge was seen, so that we may agree on a day and time to assess them in the laboratory for the muscle and pain characteristics. This has now been clarified in the text. On line 135 onwards, we clarify that following seeing the surge, we would expect ovulation within 14-26 hours, so the participant would call us to book the laboratory session (with a 2-days window allowance to fit with their availability). See section 2.3 expanded.
R1Q3: From the results, it appears that testing for the ovulatory phase occurred on day 14 for all participants however as the range of menstrual cycle length was from 23 to 45 days, it is highly likely that the ovulatory phase was missed in a number of participants. This would also help explain why there was no significant difference in hormone levels over the three phases.
R1R3: We thank our reviewer for spotting this typographical error. Ovulation day was called ‘day 14’ in our lab nomenclature but we failed to report what this day actually corresponded to for each participant. We now report the correct value which is day 16+/-4 (not day 14). We agree, the lack of hormonal differences across the 3 different days in this sample remains a puzzle. One possible explanation is that participant mis-reported their LH surge as this was unsupervised by the researcher (data collected by each participant at home). We now add this lack of supervision as a potential study limitation.
R1Q4: It is also not clear how the timing for testing of the luteal phase was determined.
R1R4: We apologise for this omission and have now expanded on the timing for the luteal and follicular phases within section 2.3.
R1Q5: There is also an issue with the title and the aim of the study to assess over the phases of the menstrual cycle since the expected hormonal fluctuations to confirm menstrual cycle phase did not occur.
R1R5: We feel that our title ‘across three time points during the menstrual cycle’ does not allude to whether the time points are different phases or not, in the menstrual cycle. For this reason, we feel that it reflects the work that was carried out and now better described, following our additional clarifications based on our reviewer’s helpful comments.
Similarly, we do not believe that it is good practice to revise our aims in view of our findings. The fact that our results are not entirely as expected does not detract from the original aims.
R1Q6: Since there were no differences in the group data, individual changes were assessed. Did the authors consider excluding the participants who did not experience the expected hormonal fluctuations (see https://pubmed.ncbi.nlm.nih.gov/31246715/#:~:text=Conclusion%3A%20To%20improve%20the%20quality,of%20serum%20estrogen%20and%20progesterone and https://pubmed.ncbi.nlm.nih.gov/33725341/ for recommendations)?
It could be that the inclusion of participants who did not experience the expected fluctuations or who were not in the correct phase at the time of testing is masking group differences. It may be worthwhile looking to see how many participants would be included based on their hormone levels and re-running the group analysis.
R1R6: We have now rephrased (at the end of the ‘Introduction’, and the start of the ‘Discussion’) our aims to make it clear that different time points and hormonal profiles were compared and contrasted.
We thank our reviewer for the suggested reading material. In this small sample of n=11, we do not believe that it would be helpful to over interpret the data as it stands or attempt p-hacking by retrospectively removing some participants to see what effect this may have on outcomes. Certainly, we feel that more work is needed in this area (including larger samples) and this is now stated in a ‘Study limitations and recommendations for future work’ section at the end of the ‘Discussion’.
Reviewer 2 Report
Comments and Suggestions for Authors
The study is very laborious and rigorously structured according to Material and method, Design, Statistical analysis, presentation of results, discussion, and conclusions. It is a complex study, an evaluation one, without any intervention on the subjects; it is based on the evaluation of 3 phases using modern methods of muscle quality evaluation (ultrasonography), correctly structured and cleanly exposed.
Some suggestions:
1. Please complete the limitations of the study, for example, the small sample of subjects.
2. You mentioned inclusion criteria, to which I would add to the absence of contraception medication that you specify the absence of other drugs or supplements ingested during the study.
3. Please mention the exclusion criteria.
4. Please provide your ClinicalTrials.gov registration number, if applicable.
Author Response
R2Q1: The study is very laborious and rigorously structured according to Material and method, Design, Statistical analysis, presentation of results, discussion, and conclusions. It is a complex study, an evaluation one, without any intervention on the subjects; it is based on the evaluation of 3 phases using modern methods of muscle quality evaluation (ultrasonography), correctly structured and cleanly exposed.
Some suggestions:
R2R1: We thank our reviewer for appreciating the sheer amount of work, coordination, planning and data analysis we carried out to produce this work.
R2Q2: 1. Please complete the limitations of the study, for example, the small sample of subjects.
R2R2: In agreement with the reviewer, we have now added a ‘Study limitations and recommendations for future work’ section at the end of the ‘Discussion’ and this refers to the sample size.
R2Q3: 2. You mentioned inclusion criteria, to which I would add to the absence of contraception medication that you specify the absence of other drugs or supplements ingested during the study.
R2R3: As advised by our reviewer, we have now added further conditions vis-à-vis the medication and supplement status of the study participants.
R2Q4: 3. Please mention the exclusion criteria.
R2R4: We have now specified the exclusion criteria.
R2Q5: 4. Please provide your ClinicalTrials.gov registration number, if applicable.
R2R5: This was not a clinical trial so we are unable to provide such a registration number.